# Rising Incidence of Non-Cardia Gastric Cancer among Young Women in the United States, 2000–2018: A Time-Trend Analysis Using the USCS Database

**DOI:** 10.3390/cancers15082283

**Published:** 2023-04-13

**Authors:** Janice Oh, Yazan Abboud, Miguel Burch, Jun Gong, Kevin Waters, Jenan Ghaith, Yi Jiang, Kenneth Park, Quin Liu, Rabindra Watson, Simon K. Lo, Srinivas Gaddam

**Affiliations:** 1Department of Medicine, Cedars-Sinai Medical Center, Los Angeles, CA 90048, USA; 2Karsh Division of Gastroenterology and Hepatology, Cedars-Sinai Medical Center, Los Angeles, CA 90048, USA

**Keywords:** SEER database, NPCR database, incidence rate trend, non-cardia gastric cancer

## Abstract

**Simple Summary:**

Although the global incidence of non-cardia gastric cancer (NCGC) is decreasing, there are limited data on sex-specific incidence in the U.S. The aim of our study was to investigate age and sex-specific time trends of NCGC incidence using a large national database from the Surveillance, Epidemiology, and End Results (SEER) Program, and thereafter to externally validate these findings within an independent population-based database, the National Program of Cancer Registries (NPCR). We evaluated the influence of race, histopathology, and disease stage at diagnosis on these trends. NCGC incidence has been relatively increasing at a greater rate in younger women compared to counterpart men over the last two decades, and this rise may be explained by a disproportionate increase in young non-Hispanic White women. Our findings support the increasing incidence of cancers at multiple sites in young women, and emphasize the need for dedicated research to determine the etiologies of these alarming trends.

**Abstract:**

Introduction: Although the global incidence of non-cardia gastric cancer (NCGC) is decreasing, there are limited data on sex-specific incidence in the United States. This study aimed to investigate time trends of NCGC from the SEER database to externally validate findings in a SEER-independent national database, and to further assess trends among subpopulations. Methods: Age-adjusted incidence rates of NCGC were obtained from the SEER database from 2000 to 2018. We used joinpoint models to calculate average annual percentage change (AAPC) to determine sex-specific trends among older (≥55 years) and younger adults (15–54 years). Using the same methodology, findings were then externally validated using SEER-independent data from the National Program of Cancer Registries (NPCR). Stratified analyses by race, histopathology, and staging at diagnosis were also conducted in younger adults. Results: Overall, there were 169,828 diagnoses of NCGC from both independent databases during the period 2000–2018. In SEER, among those <55 years, incidence increased at a higher rate in women (AAPC = 3.22%, *p* < 0.01) than men (AAPC = 1.51%, *p* = 0.03), with non-parallel trends (*p* = 0.02), while a decreasing trend was seen in both men (AAPC = −2.16%, *p* < 0.01) and women (AAPC = −1.37%, *p* < 0.01) of the ≥55 years group. Validation analysis of the SEER-independent NPCR database from 2001 to 2018 showed similar findings. Further stratified analyses showed that incidence is disproportionately increasing in young non-Hispanic White women [AAPC = 2.28%, *p* < 0.01] while remaining stable in their counterpart men [AAPC = 0.58%, *p* = 0.24] with non-parallel trends (*p* = 0.04). This pattern was not observed in other race groups. Conclusion: NCGC incidence has been increasing at a greater rate in younger women compared to counterpart men. This disproportionate increase was mainly seen in young non-Hispanic White women. Future studies should investigate the etiologies of these trends.

## 1. Introduction

Over one million new cases of gastric cancer were reported worldwide in 2020 and it is the fourth leading cause of cancer mortality [1]. By 2022, the incidence of gastric cancer in the U.S. was estimated to be over 26,000 cases, higher than that of esophageal, small intestine, or biliary cancer [2]. The 5-year survival rate in the U.S. is estimated to be about 32.4% [3], which significantly decreases to 6% in advanced metastatic stages [4]. With disease progression, patients experience deteriorating symptoms that cause significant psychological distress [5]. Furthermore, with its high recurrence rate of up to 60% in the recent literature [6], healthcare for surveillance, maintenance, and treatment of gastric cancer is costly and resource-intensive [7].

Of the two anatomical subtypes of gastric cancer, non-cardia gastric cancer (NCGC) is associated with different pathophysiology and risk factors than cardia gastric cancer, which behaves like, and is frequently diagnosed as, esophageal adenocarcinoma [8]. Thus, to understand the burden of gastric cancer alone, it is important to study the epidemiology of primary NCGC. A study using the 45 North American Association of Central Cancer Tumor Registries demonstrated an increasing trend of NCGC incidence in younger individuals, with the trend being more pronounced in women than men [9]. However, age and sex-specific incidence trends of subpopulations, such as in younger non-Hispanic White women, are currently limited in the literature. With a recent rise in gastrointestinal cancers in younger individuals, it is crucial to quantify subgroup risks using large population databases to uncover age and sex-specific trends that may be masked, especially in young individuals.

The overall objective of this study was to assess the incidence trends of NCGC in a nationally representative sample. We aimed to investigate age and sex-specific time trends of NCGC incidence in the U.S. using a large database from the Surveillance, Epidemiology, and End Results (SEER) Program, and thereafter to externally validate these findings within an independent population-based database, the National Program of Cancer Registries (SEER-independent NPCR). Secondarily, we also evaluated the influence of risk factors including race, histopathology, and disease stage at diagnosis on these age and sex-specific trends using the SEER database.

## 2. Methods

To evaluate the time trends of NCGC incidence rates in the U.S., we obtained data from 1 January 2000 to 31 December 2018 from the SEER database. We used data from 1 January 2001 to 31 December 2018 from the NPCR database for external validation. Both databases are publicly available with de-identified patient information, and therefore this study was exempt from IRB review based on our institutional policy.

### 2.1. Databases and Exclusion Criteria

The SEER 21 Program has been funded by the National Cancer Institute (NCI) to collect cancer statistics from 19 geographic regions, encompassing approximately 34.6% of the U.S. population [10]. Alternatively, the NPCR is funded by the Centers for Disease Control and Prevention (CDC) to collect data from cancer registries in 46 states, the District of Columbia, Puerto Rico, the U.S. Pacific Island Jurisdictions, and the U.S. Virgin Islands [10]. We used SEER data in their entirety, including all state and regional registries. For the steps of external validation, we excluded state cancer registries from the NPCR database that also reported data in part or in full to SEER, resulting in the ‘SEER-independent NPCR database’. The states that were excluded were Alaska, California, Connecticut, Georgia, Hawaii, Idaho, Iowa, Kentucky, Louisiana, Massachusetts, New Jersey, New Mexico, New York, Utah, and Washington [8]. The ‘SEER-independent NPCR database’ consisted of cancer statistics from 35 states and Washington D.C., covering nearly 64.5% of the U.S. population (Appendix A) [11]. Thus, this study evaluated the NCGC incidence trends from two nationally representative databases that collectively cover approximately 100% of the U.S. population. Data were obtained from the SEER and NCPR databases using SEER*Stat software, v8.4.0.1 (NCI, Bethesda, MD, USA). 

NCGC was defined as cancer in all regions of the stomach, except for the cardia. We excluded data points with any cardia involvement, including cardia, overlapping, and unspecified subsites (*ICD-10* codes 16.0, 16.8, 16.9). Only cancers of non-cardia gastric origin were included in the study, and cases of leukemia, lymphoma, mesothelioma, or Kaposi’s sarcoma cancers (*ICD-O-3* codes 9050-9055, 9140, 9590-9989) were excluded from all analyses [12].

### 2.2. Statistical Analysis

The incidence rates were adjusted to reporting delay and age to the 2000 US population, and the age-standardized incidence rate (aIR) was defined as the number of individuals diagnosed with NCGC per 100,000 population per year. We conducted a time-trend analysis of the quantified rates using the Joinpoint Regression Program, version 4.9.1 (NCI, Bethesda, MD, USA). This software program performs the Monte Carlo permutation method to determine the simplest joinpoint model that best fits the time trend of the NCGC incidence rates from the SEER and SEER-independent NPCR datasets [13]. Following the fitting of the model, the program quantifies the rate trends in annual percentage changes (APC) with its average (AAPC), and tests the statistical difference from zero using a two-tailed test (*p*-value < 0.05) based on asymptotic normality and the calculated variances of the estimated joinpoint segments [14]. The trends were considered stable if AAPC *p*-values were non-significant. Increasing or decreasing trends had AAPC values that were positive or negative, respectively. Furthermore, pairwise comparison tests are conducted to compare the sex-specific trends for parallelism and identicalness, using the test of parallelism and test of coincidence, respectively, under the assumption of uncorrelated errors with constant variance [15]. The test of parallelism assessed if the incidence trends were parallel based on the AAPCs of the linear joinpoint segments. A statistically significant result indicated a difference between the estimated slopes of the two independent comparison groups, and *p*-values < 0.05 signified that the two time trends were neither parallel nor identical. Subsequently, we evaluated trends by risk factors with a cutoff age at 55 years to determine trends among older (≥55 years) and younger adults (15–54 years). The trends were considered significant using a 2-tailed *t*-test (*p*-value < 0.05). Thereafter, the same analysis was performed in the SEER-independent NPCR database for external validation. 

Lastly, to distinctively investigate trends in younger adults, analyses of sex-specific trends were further stratified by race, histopathology, and staging at diagnosis. Self-reported race and ethnicity data were extracted from medical records and categorized into non-Hispanic White, non-Hispanic Black, Hispanic, Asian, and others [16]. Lauren’s criteria were applied to classify cases of NCGC into histology groups of intestinal, diffuse, and other (unspecified) subtypes [8]. *ICD-O-3* codes for each subtype group were established on the basis of prior studies [9,12] and included the most common histopathology diagnoses. Staging at diagnosis was grouped into localized, regional, distant, and unknown/unstaged. Of note, the data for staging at diagnosis are from 2004 to 2018, as the staging collection and reporting process in SEER changed for all registries in 2004 [11]. Given that the race grouping options were limited in the SEER-independent NPCR database, we assessed these trends using the SEER database only. 

## 3. Results

### 3.1. Discovery Phase: SEER Database 2000–2018

#### 3.1.1. Overall Rates and Trends

According to the SEER database, a total of 79,068 patients were diagnosed with NCGC during the period 2000–2018. The overall aIR of NCGC in men (4.44, 95% CI 4.39–4.48) was higher than in women (2.91, 95% CI 2.88–2.94) (Table 1). As seen in Table 2A, the overall trends were significantly decreasing in both men [AAPC = −1.73% (−1.94–−1.51%); *p* < 0.01] and women [AAPC = −0.60% (−0.80–−0.40%); *p* < 0.01]. The sex-specific trends were both non-parallel and non-identical (both *p* < 0.01), suggesting that incidence rates in men and women were both decreasing but at different rates. 

#### 3.1.2. Sex-Specific Trends by Age Groups

Among those aged ≥55 years, trends were decreasing in both men [AAPC = −2.16% (−2.38–−1.95%); *p* < 0.01] and women [AAPC = −1.37% (−1.61–−1.12%); *p* < 0.01] (Figure 1A). The sex-specific trends were neither parallel nor identical (both *p* < 0.01). 

Of those younger than 55 years (15–54), a total of 13,791 individuals (47.7% women) were diagnosed with NCGC during 2000–2018. The overall aIR of NCGC in men (1.21, 95%CI 1.18–1.24) was higher than in women (1.09, 95%CI 1.06–1.11). Overall trends of the aIRs significantly increased in both men [AAPC = 0.68% (0.06–1.30%); *p* = 0.03] and women [AAPC = 2.28% (1.82–2.74); *p* < 0.01]. However, as seen in Figure 1B, the NCGC incidence rate in women surpassed that in men after 2016. The two sex-specific trends were neither parallel (*p* = 0.02) nor identical (*p* < 0.01). 

### 3.2. External Validation Phase: SEER-Independent NPCR Database 2001–2018

#### 3.2.1. Overall Rates and Trends

After the exclusion of 78,757 individuals that were also reported to the SEER Program, a total of 90,760 patients were diagnosed with NCGC from 2001 to 2018 in the SEER-independent NPCR database. The overall incidence rate of NCGC in men (3.06, 95%CI 3.03–3.09) was also higher than in women (2.03, 95%CI 2.01–2.05). Notably, the mean trend was significantly decreasing in men [AAPC = −1.55% (−2.37–−0.73%); *p* < 0.01], while remaining stable in women [AAPC = −0.19% (−0.98–−0.61%); *p* = 0.65]. The sex-specific trends were both non-parallel and non-identical (both *p* < 0.01; Table 2B). 

#### 3.2.2. Sex-Specific Trends by Age Groups

As demonstrated in Figure 1C, among men and women of ages ≥55 years, mean sex-specific trends of this subgroup were decreasing in both men [AAPC = −2.06% (−3.01–−1.11%); *p* < 0.01] and women [AAPC = −0.82% (−1.22–−0.43%); *p* < 0.01]. The trends were non-parallel and non-identical (both *p* < 0.01). 

Of those younger than 55 years (15–54), a total of 15,486 individuals (47.4% women) were diagnosed with NCGC during 2001–2018. The overall sex-specific incidence rate of NCGC in men (0.88, 95%CI 0.86–0.90) was higher than in women (0.79, 95%CI 0.77–0.81). The incidence increased in both men [AAPC = 1.51% (1.04–1.98%); *p* < 0.01] and women [AAPC = 3.22% (2.69–3.77%); *p* < 0.01]. As also seen in Figure 1D, the NCGC incidence rate in women surpassed that in men after 2016.The two sex-specific trends were neither parallel nor identical (both *p* < 0.01). 

### 3.3. Evaluation of Risk Factors among Young Adults

#### 3.3.1. Sex-Specific Trends by Race/Ethnicity

The overall incidence rates were higher in non-Hispanic Blacks (1.80, 95%CI 1.73–1.87), Hispanics (1.97, 95%CI 1.91–2.03), and Asians/Pacific Islanders (2.02, 95%CI 1.94–2.11) compared to non-Hispanic Whites (0.62, 95%CI 0.60–0.64), seen in both men and women (Table 1). The time trends were parallel among men and women in non-Hispanic Blacks, Hispanics, and Asians/Pacific Islanders (all *p* > 0.05). Notably, despite lower incidence rates in both sexes, the incidence trend in younger non-Hispanic White adults was increasing in women [AAPC = 2.28% (1.38–3.19%); *p* < 0.01] while stable in their counterpart men [AAPC = 0.58% (−0.42–1.59%); *p* = 0.24] (Figure 2). Further, the sex-specific trends in younger non-Hispanic Whites were neither parallel (*p* = 0.04) nor identical (*p* < 0.01). 

#### 3.3.2. Sex-Specific Trends by Histopathology

Based on Lauren’s classification, there were more cases of diffuse or unspecified subtypes of NCGC (5280 total diffuse; 7010 total unspecified) than the intestinal subtype (1364 total intestinal) from 2000 to 2018 in the younger population (Table 3). When stratified by subtypes, incidence rates were lower in women than men for intestinal (aIRR 0.53) and unspecified subtypes (aIRR 0.87), while higher among the diffuse subtype (aIRR 1.07). Both sex groups had an increasing incidence of intestinal and diffuse subtypes of NCGC, with parallel trends (both *p* > 0.05) (Figure 3 and Appendix A). Notably, of the unspecified subtype, women had a statistically significant increase in the incidence trend [AAPC = 3.62% (2.99–4.26%) *p* < 0.01] while the trends in men were stable [AAPC = 0.43% (−0.40–1.21%); *p* = 0.26]. Both sex-specific trends were neither parallel nor identical (both *p* < 0.01). 

#### 3.3.3. Sex-Specific Trends by Staging at Diagnosis

When stratified by staging at diagnosis, women had higher overall incidence than men in localized stages during the time period 2004–2018. On the contrary, overall incidence rates in men were higher in regional and distant stages than in women. Notably, in those diagnosed with localized primary NCGC, the incidence trend was increasing in women [AAPC = 5.17% (4.04–6.31%); *p* < 0.01] at a greater rate than in men [AAPC = 2.04% (0.33–3.78%); *p* = 0.02]. The trends were neither parallel nor identical (both *p* = 0.03). As seen in Figure 4, the time trends in both sexes were parallel in regional, distant, and unknown subgroups (all *p* > 0.05).

## 4. Discussion

This study is among the largest and most updated population-based analyses evaluating the incidence time trends of primary NCGC in the U.S. from 2000 to 2018. We found that among 79,068 individuals diagnosed with NCGC from the SEER database, the overall rates have been decreasing in the U.S for both men and women. However, when stratified by sex and age, the incidence rates are increasing in younger women (15–54 years) at a greater rate than in young men, with women surpassing men in rates after 2016. Our external validation phase using the SEER-independent NPCR dataset of 90,760 individuals found similar results of the incidence trends. 

Despite similar findings, there were small differences in the analysis results. Joinpoints are included in the regression models of SEER-independent NPCR data, specifically in both sexes of ‘all ages’ and in the subgroup of men ≥55 years of age. These slight differences between datasets may possibly be due to variation in sample sizes, as more significant trend changes can be detected with larger sample sizes [17]. We can also consider geographical differences in data collection for each database that may influence demographic and environmental factors that correlate to NCGC incidence. Nonetheless, the primary conclusion from each analysis of the two independent databases is the same: the overall NCGC incidence rate appears to be decreasing over the past two decades while increasing at a greater rate in young women compared to their counterpart men. 

In the further evaluation of risk factors in NCGC, there were no statistically significant differences in incidence trends between younger men and women with intestinal or diffuse histopathology subtypes. In contrast, among those with an unspecified subtype, women had a markedly increasing incidence rate of NCGC from 2000 to 2018 compared to their counterpart men. Although this may suggest an underlying sex discrepancy in NCGC incidence by histopathology, it is difficult to make definitive conclusions due to limitations of incomplete data reporting and lack of heterogeneity in unspecified histology subtypes, as more than two-thirds consisted of undifferentiated adenocarcinoma. When stratified by staging at diagnosis, the incidence trend of localized primary NCGC was increasing in women at a greater rate than in men during 2004–2018. Meanwhile, time trends in both sexes were parallel in regional, distant, and unknown subgroups (all *p* > 0.05). The reasons for the discrepancy seen in localized stages are unknown, although the *H. pylori* infection mechanism and epigenetics that influence the pathophysiology of NCGC cellular spread may be possibilities.

We also found that among young adults, despite higher incidence rates in both sexes of Hispanics, non-Hispanic Blacks, and Asians, the trend of incidence over the past two decades is markedly increasing in non-Hispanic White women while remaining stable in non-Hispanic White men. Based on our APC comparisons, primary NCGC incidence in young non-Hispanic White women has been increasing at a rate nearly four times that of their counterpart men over the last two decades. The disproportionate increase is consistent with findings by Anderson et al.; however, our study quantifies the discrepancy between the focused cohorts using more recent data. It is also important to note that the increase in NCGC incidence was statistically significant in both young Hispanic men and women with parallel time trends, differing from the conclusions of studies that have demonstrated a higher increasing rate in one sex than another [18,19]. Despite the inconsistent findings in the recent literature, further evaluation of this trend among young Hispanics is warranted.

Many studies in the existing literature have investigated the incidence of gastric cancer and its associated risk factors in global settings. It is widely accepted that *H. Pylori* infections predispose individuals to NCGC [20], and *H. pylori* transmission dynamics may be playing a role in the increasing NCGC incidence in young women. Prior studies demonstrate that *H. pylori* is a strong risk factor for gastric cancer in younger non-Hispanic White women, in whom H. pylori infection prevalence had been previously low, making them new, vulnerable hosts [21]. A review by Thrift et al. also explores risk factors for developing gastric cancer from *H. pylori* infections, such as reduction in gastric acid secretion from oxyntic atrophy of the gastric fundus or body, and different carcinogenic potential based on various strains and virulence factors such as cytotoxin-associated-gene A (cagA) [22]. Thus, future studies can be guided to assess the incidence of cagA+ strain infections in young women compared to men or older women, and to determine any differences in oxyntic atrophy and gastric acidity levels among various population subgroups. More importantly, additional research entailing cost-effectiveness analysis in *H. pylori* screening strategies in young women is warranted.

In addition to *H. pylori* infections, the increasing NCGC incidence trend in young non-Hispanic White women can be attributed to changes in gut microbiota [23], differences in hormone exposure [24], and effects from tobacco and alcohol use [25]. Additionally, genetic factors seem to affect females to a greater degree than males [26,27], suggesting that exposure risks and epigenetic processes may be contributing to the increase in incidence. The current literature, from a genetic standpoint, suggests a higher risk for gastric cancer from microsatellite instability, which is more associated with women but of older age (>65 years) [28]. Furthermore, it can be hypothesized that the disproportionate increase in NCGC incidence trend in younger women is associated with genetically-predisposed autoimmunity. Notably, autoimmune gastritis (AIG) is three times more common in women than men [29]. A recent cohort study found that antibodies of AIG were associated with increased gastric cancer risk, being strongest in non-cardia locations [30]. There was also speculation of more antibiotics and proton-pump inhibitor (PPI) therapy use in women than men [31], which alter the gut microbiome potentially eliciting an inflammatory response. Altogether, the higher risk for autoimmunity and medication-induced inflammatory states in young women than men may play a role in the sex-specific differences in primary NCGC incidence seen in this study.

There are strengths and limitations to this study. A notable strength is that this study is the largest population-based study to date which evaluates NCGC incidence time trends in the U.S., covering nearly 100% of the U.S. population. The use of two independent cohorts with 2018 data offers more accurate estimates of the current incidence trend. Secondly, data extracted from the SEER database were adjusted for reporting delay, which is important in precisely determining updated trends [32]. Lastly, this study conducted external validation to demonstrate the reproducibility of its findings. One limitation is that there may be a small overlap between the SEER and SEER-independent NPCR databases, despite the exclusion of registries. However, we removed more than 79,000 data points with the state-wide exclusion method, and the small overlap is estimated to be approximately less than 1%. Another limitation of this study was the smaller sample sizes of the younger adults when using only the SEER database for the secondary analyses instead of both databases combined. Due to limited race grouping availabilities for the NPCR database, we opted to use only the SEER database to evaluate the incidence trends among more focused race subgroups. Moreover, it is unclear if race and ethnicity self-identification represents ancestry or whether it is culture-based, especially in individuals of mixed races. As initial race and ethnicity data are charted by the providers, there may be some inconsistency in assignment and coding as there is no standardized collection method used across all facilities. 

## 5. Conclusions

Based on two independent population databases to cover nearly 100% of the U.S. population, NCGC incidence rates were relatively increasing in younger women compared to younger men over the last two decades. Furthermore, there is an increasing trend in young non-Hispanic White women compared to counterpart men. Our findings support the increasing incidence of cancers at multiple sites in young women [33], although the reasons for these alarming trends remain elusive. Thus, future studies should investigate linked or separate etiologies of these trends to decrease the overall burden of primary NCGC in the future with newly developed screening or preventive measures. 

## Figures and Tables

**Figure 1 cancers-15-02283-f001:**
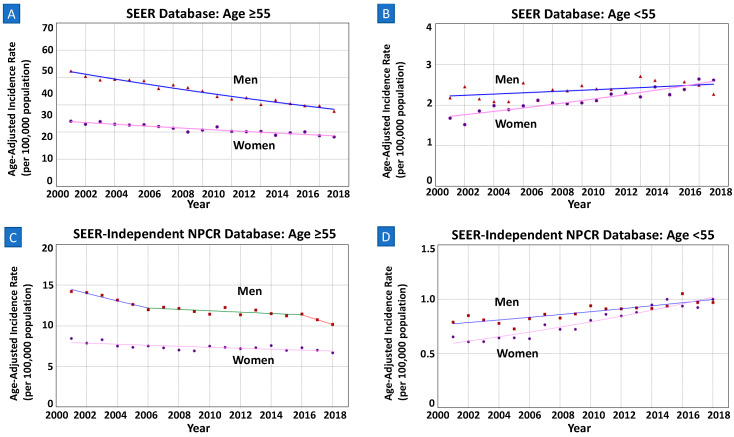
(**A**) In SEER, among those aged ≥55 years, APC in men is decreasing at a greater rate than in women with non-parallel trends (AAPC −2.44% vs. −1.64%; *p*-value < 0.01). (**B**) In SEER, among those aged <55 years, incidence increased in women but remained stable in men (AAPC 2.01% vs. 0.29%; *p*-value < 0.01). (**C**,**D**) In SEER-Independent NPCR, both findings in those of age ≥55 and <55 were similar to the analyses of SEER database.

**Figure 2 cancers-15-02283-f002:**
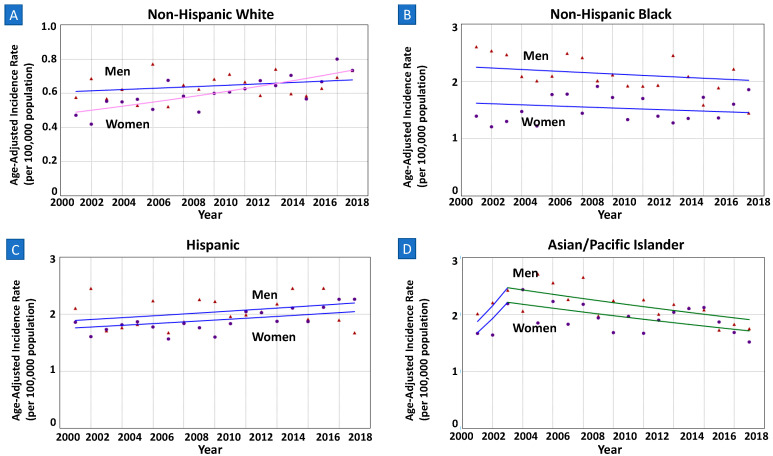
(**A**–**D**) Sex-specific incidence trends among younger adults by race. All data are from SEER database, among those of age < 55 years. Lines of both men and women are same colors if trends are parallel, meaning same APC and AAPC values during the interval years. (**A**) In Non-Hispanic Whites, APC in women is increasing at a greater rate than in men with non-parallel trends (AAPC 2.28% vs. 0.58%; *p*-value < 0.04). (**B**–**D**) In Non-Hispnanic Blacks, Hispanics, and Asian/Pacific Islanders, incidence trends in women and men were identical but not parallel (*p*-values 0.06, 0.17, 0.32, respectively).

**Figure 3 cancers-15-02283-f003:**
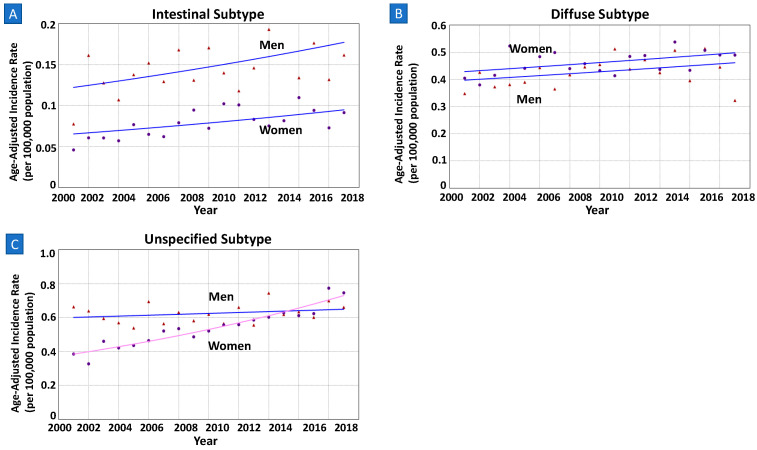
(**A**–**C**) Sex-specific incidence trends among younger adults by histopathology based on Lauren’s criteria. All data are from SEER database, among those of age < 55 years. Lines of both men and women are same colors if trends are parallel, meaning same APC and AAPC values during the interval years. (**A**,**B**) Based on Lauren’s criteria, both intestinal and diffuse subtype groups showed parallel incidence trends between men and women (*p*-values 0.29 and 0.83, respectively). (**C**) Of the unspecified subtype, women had statistically significant increase in incidence compared to counterpart men in nonparallel trend (AAPC 3.62% vs. 0.43%; *p*-value < 0.01).

**Figure 4 cancers-15-02283-f004:**
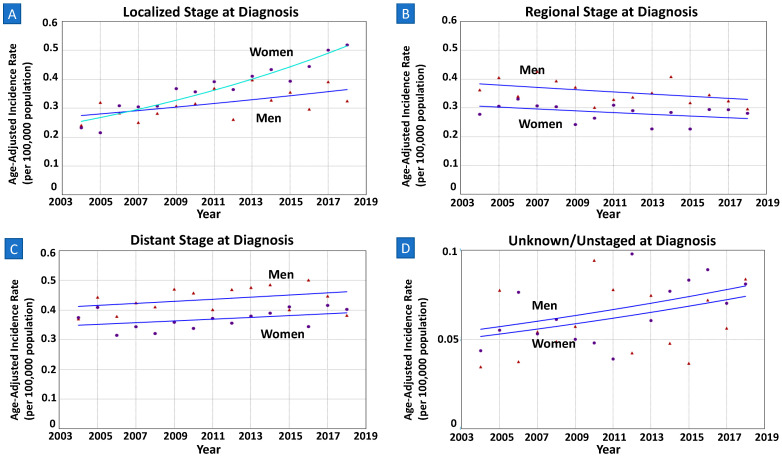
(**A**–**D**) Sex-specific incidence trends among younger adults by staging at diagnosis. All data are from SEER database, among those of age < 55 years. Lines of both men and women are same colors if trends are parallel, meaning same APC and AAPC values during the interval years. (**A**) Among patients with localized stage of disease at diagnosis, women had higher increase in incidence than men (AAPC 5.17% vs. 2.04%; *p*-value < 0.03). (**B**–**D**) Among those with regional, distant, or unknown/unstaged staging at diagnosis, the trends between men and women were parallel (*p*-values 0.47, 0.94, 0.37, respectively).

**Table 1 cancers-15-02283-t001:** Overall Incidence Rates (per 100,000 population) of Primary Non-cardia Gastric Cancer in the U.S.

SEER Database 2000–2018 (*N* = 79,068)				
Age	Overall	Women	Men	aIRR *
All Ages	3.56 (3.54–3.59)	2.91 (2.88–2.94)	4.44 (4.39–4.48)	0.66
≥55	13.59 (13.49–13.70)	10.71 (10.59–10.84)	17.54 (17.35–17.72)	0.61
<55	1.15 (1.13–1.16)	1.09 (1.06–1.11)	1.21 (1.18–1.24)	0.90
Age < 55 years (*N* = 13,791)				
Race/Ethnicity	Overall	Women	Men	aIRR *
Non-Hispanic White	0.62 (0.60–0.64)	0.60 (0.57–0.62)	0.64 (0.61–0.67)	0.94
Non-Hispanic Black	1.80 (1.73–1.87)	1.52 (1.43–1.61)	2.11 (2.00–2.23)	0.72
Hispanic	1.97 (1.91–2.03)	1.90 (1.82–1.99)	2.04 (1.95–2.13)	0.93
Asian/Pacific Islander	2.02 (1.94–2.11)	1.91 (1.81–2.03)	2.15 (2.02–2.27)	0.89
Other	1.59 (1.33–1.89)	1.40 (1.06–1.81)	1.80 (1.41–2.26)	0.78
Histopathology				
Intestinal Subtype ^a^	0.11 (0.11–0.12)	0.08 (0.07–0.09)	0.15 (0.14–0.16)	0.53
Diffuse Subtype ^b^	0.44 (0.43–0.45)	0.46 (0.44–0.48)	0.43 (0.41–0.44)	1.07
Other/Unspecified ^c^	0.58 (0.57–0.59)	0.54 (0.52–0.56)	0.62 (0.60–0.64)	0.87
Stage at Diagnosis (2004–2018)				
Localized	0.34 (0.33–0.35)	0.37 (0.35–0.39)	0.31 (0.30–0.33)	1.19
Regional	0.32 (0.31–0.33)	0.28 (0.27–0.30)	0.35 (0.34–0.37)	0.80
Distant	0.40 (0.39–0.41)	0.37 (0.35–0.39)	0.43 (0.42–0.45)	0.86
Unstaged/Unknown	0.06 (0.06–0.07)	0.07 (0.06–0.07)	0.06 (0.05–0.07)	1.17
**SEER-Independent NPCR Database 2001–2018 (*N* = 90,760)**				
Age Groups	Overall	Women	Men	aIRR *
All Ages	2.47 (2.46–2.49)	2.03 (2.01–2.05)	3.06 (3.03–3.09)	0.66
≥55	9.45 (9.27–9.40)	7.37 (7.29–7.45)	11.97 (11.85–12.08)	0.62
<55	0.83 (0.82–0.85)	0.79 (0.77–0.81)	0.88 (0.86–0.90)	0.90

Age and sex-specific NCGC incidence rates and women to men incidence rate ratios (aIRR) from SEER database, and SEER-Independent NPCR Database. Data are presented as aIR (95% CI). aIRR *: adjusted incidence rate ratio (women: men). ^a^: ICD-O-3 codes: 8012, 8021, 8022, 8031, 8032, 8046, 8050, 8082, 8143, 8144, 8201, 8210, 8211, 8220, 8221, 8255, 8260, 8261, 8262, 8263, 8310, 8323, 8480, 8481, 8510, 8512, 8570, and 8576. ^b^: ICD-O-3 codes: 8020, 8041, 8044, 8141, 8142, 8145, 8490, and 8806. ^c^: ICD-O-3 codes: 8000, 8010, 8140, 8210, 8240, 8246, 8249, 8890, and 8936.

**Table 2 cancers-15-02283-t002:** (**A**) SEER Database (2000–2018): Age and Sex-Specific Trends of Non-cardia Gastric Cancer in the United States. (**B**) SEER-Excluded NPCR Database (2001–2018): Age and Sex-Specific Trends of Non-cardia Gastric Cancer in the United States.

**(A)**							
		**Trends**				**Pairwise Comparison**	
**Sex**	**Cases** ***n* (% of Age Group)**	**Years**	**APC** **(95%CI)**	**AAPC ^a^** **(95%CI)**	**AAPC** ***p*-Value ^b^**	**Test of Coincidence** ***p*-Value ^c^**	**Test of Parallelism** ***p*-Value ^d^**
All Ages (*N* = 79,068)							
Men	42,965 (54.3%)	2000–2018	−1.73% (−1.94–−1.51)	−1.73% (−1.94–−1.51)	<0.001	<0.001	<0.001
Women	36,103 (45.7%)	2000–2018	−0.60% (−0.80–−0.40)	−0.60% (−0.80–−0.40)	<0.001		
Age ≥ 55 (*N* = 65,258)							
Men	35,747 (54.8%)	2000–2018	−2.16% (−2.38–−1.95)	−2.16% (−2.38–−1.95)	<0.001	<0.001	<0.001
Women	29,511 (45.2%)	2000–2018	−1.37% (−1.61–−1.12)	−1.37% (−1.61–−1.12)	<0.001		
Age < 55 (*N* = 13,791)							
Men	7212 (52.3%)	2000–2018	0.68% (0.06–1.30)	0.68% (0.06–1.30)	0.033	<0.001	0.015
Women	6579 (47.7%)	2000–2018	2.28% (1.82–2.74)	2.28% (1.82–2.74)	<0.001		
**(B)**							
		**Trends**				**Pairwise Comparison**	
**Sex**	**Cases** ***n* (% of Age Group)**	**Years**	**APC** **(95%CI)**	**AAPC ^a^** **(95%CI)**	**AAPC** ***p*-Value ^b^**	**Test of Coincidence** ***p*-Value ^c^**	**Test of Parallelism** ***p*-Value ^d^**
All Ages (*N* = 90,760)							
Men	49,740 (54.8%)	2001–2006	−2.92% (−4.42–−1.39)	−1.55% (−2.37–−0.73)	<0.001	<0.001	<0.001
		2006–2016	−0.23% (−0.83–0.37)				
		2016–2018	−4.61% (−10.50–1.66)				
Women	41,020 (45.2%)	2001–2005	−2.36% (−5.55–0.94)	−0.19% (−0.98–−0.61)	0.65		
		2005–2018	0.49% (−0.04–1.02)				
Age ≥ 55 (*N* = 75,241)							
Men	41,585 (55.3%)	2001–2006	−3.39% (−5.11–1.63)	−2.06% (−3.01–−1.11)	<0.001	<0.001	0.009
		2006–2016	−0.72% (−1.41–−0.02)				
		2016–2018	−5.39% (−12.12–1.87)				
Women	33,656 (44.7%)	2001–2018	−0.82% (−1.22–−0.43)	−0.82% (−1.22–−0.43)	<0.001		
Age < 55 (*N* = 15,486)							
Men	8144 (52.6%)	2001–2018	1.51% (1.04–1.98)	1.51% (1.04–1.98)	<0.001	<0.001	<0.001
Women	7342 (47.4%)	2001–2018	3.22% (2.69–3.77)	3.22% (2.69–3.77)	<0.001		

^a^: AAPC is calculated as average of APCs over the designated time period. ^b^: For all ages, AAPCs with *p*-values < 0.05 were considered significant. ^c^: Tests whether sex-specific trends were identical. *p*-value < 0.05 signifies trends were not identical. ^d^: Tests whether sex-specific trends were equal and parallel. *p*-value < 0.05 signifies the trends were not equal.

**Table 3 cancers-15-02283-t003:** Age and Sex-Specific Trends of Non-cardia Gastric Cancer Among Race and Histology Subgroups in Those of Ages < 55 Years, SEER Database 2000–2018 (*N* = 13,791).

			Trends				Pairwise Comparison	
Subgroup	Sex	Cases *n* (% of Age Group)	Years	APC (95%CI)	AAPC ^a^ (95%CI)	AAPC *p*-Value ^b^	Test of Coincidence *p*-Value ^c^	Test of Parallelism *p*-Value ^d^
Race								
Non-Hispanic White	Male	2375 (17.2%)	2000–2018	0.58% (−0.42–1.59)	0.58% (−0.42–1.59)	0.082	<0.001	0.042
	Female	2169 (15.7%)	2000–2018	2.28% (1.38–3.19)	2.28% (1.38–3.19)	0.005		
Non-Hispanic Black	Male	1395 (10.1%)	2000–2018	−0.59% (−1.49–0.31)	−0.59% (−1.49–0.31)	0.19	<0.001	0.062
	Female	1129 (8.2%)	2000–2018					
Hispanic	Male	2172 (15.7%)	2000–2018	0.84% (0.14–1.53)	0.84% (0.14–1.53)	0.020	<0.001	0.17
	Female	2027 (14.7%)	2000–2018					
Asian/Pacific Islander	Male	1170 (8.5%)	2000–2002	14.73% (−10.72–47.42)	0.10% (−2.63–2.91)	0.94	<0.001	0.32
			2002–2018	−1.59% (−2.39–−0.79)				
	Female	1173 (8.5%)	2000–2002	14.73% (−10.72–47.42)			<0.001	0.32
			2002–2018	−1.59% (−2.39–−0.79)				
Other/Unspecified	Male	72 (0.01%)	2000–2018	Could not analyze			<0.001	-
	Female	57 (0.004%)	2000–2018	Could not analyze				
Histology								
Intestinal ^e^	Male	886 (6.4%)	2000–2018	2.09% (0.95–3.24)	2.09% (0.95–3.24)	0.001	<0.001	0.29
	Female	478 (3.5%)	2000–2018					
Diffuse ^f^	Male	2514 (18.2%)	2000–2018	0.84% (0.20–1.47)	0.84% (0.20–1.47)	0.011	0.11	0.83
	Female	2766 (20.1%)	2000–2018					
Other/Unspecified ^g^	Male	3736 (27.1%)	2000–2018	0.43% (−0.4–1.21)	0.43% (−0.4–1.21)	0.26	<0.001	<0.001
	Female	3274 (23.7%)	2000–2018	3.62% (2.99–4.26)	3.62% (2.99–4.26)	<0.001		
Stage at Diagnosis								
Localized	Male	1503 (10.9%)	2004–2018	2.04% (0.33–3.78)	2.04% (0.33–3.78)	0.023	0.004	0.025
	Female	1780 (12.9%)	2004–2018	5.17% (4.04–6.31)	5.17% (4.04–6.31)	<0.001		
Regional	Male	1692 (12.3%)	2004–2018	−1.08% (−1.94–−0.20)	−1.08% (−1.94–−0.20)	0.018	<0.001	0.47
	Female	1352 (9.8%)	2004–2018					
Distant	Male	2050 (14.9%)	2004–2018	0.81% (0.04–1.58)	0.81% (0.04–1.58)	0.039	0.001	0.94
	Female	1756 (12.7%)	2004–2018					
Unknown/Unstaged	Male	288 (2.1%)	2004–2018	2.61% (0.20–5.08)	2.61% (0.20–5.08)	0.035	0.58	0.36
	Female	317 (2.3%)	2004–2018					

^a^: AAPC is calculated as average of APCs over the designated time period. ^b^: For all ages, AAPCs with *p*-values < 0.05 were considered significant. ^c^: Tests whether sex-specific trends were identical. *p*-value < 0.05 signifies trends were not identical. ^d^: Tests whether sex-specific trends were equal and parallel. *p*-value < 0.05 signifies the trends were not equal. ^e^: *ICD-O-3* codes: 8012, 8021, 8022, 8031, 8032, 8046, 8050, 8082, 8143, 8144, 8201, 8210, 8211, 8220, 8221, 8255, 8260, 8261, 8262, 8263, 8310, 8323, 8480, 8481, 8510, 8512, 8570, and 8576. ^f^: *ICD-O-3* codes: 8020, 8041, 8044, 8141, 8142, 8145, 8490, and 8806. ^g^: *ICD-O-3* codes: 8000, 8010, 8140, 8210, 8240, 8246, 8249, 8890, and 8936.

## Data Availability

All primary non-cardia gastric cancer incidence data used in this study are openly available from the CDC’s NPCR and NCI’s SEER program, which can be accessed through the SEER*Stat software (v8.4.0.1).

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
