# Peer review of "Rising Incidence of Non-Cardia Gastric Cancer among Young Women in the United States, 2000–2018: A Time-Trend Analysis Using the USCS Database"

_cancers, 2023, doi:10.3390/cancers15082283_

Round 1

Reviewer 1 Report

Reviewer really appreciates having an opportunity to evaluate this manuscript titled “Rising Incidence of Non-Cardia Gastric Cancer Among Young Women in the United States, 2000-2018: A Time-Trend Analysis Using the USCS Database” written by Dr. Oh et al. Reviewer understood that NCGC incidence has been increasing at a greater rate in younger women compared to counterpart men and this disproportionate increase was mainly seen in young non-Hispanic White women, while reviewer does not understand how the data is useful in treatment for gastric cancer. However, such a type of knowledge regarding gastric cancer are important to understand what gastric cancer is in the epidemiology. Reviewer learned many things reading this manuscript. This manuscript is valuable to be published in this journal in a proper section.

 Very minor comment

The structural abstract is divided into Introduction, Methods, Results and Discussion. Is that true? Reviewer would say that Discussion may be Conclusions. If the authors agree with reviewer’s opinion, reviewer would like the authors to correct it.

Reviewer 2 Report

This manuscript aims to investigate the sex-specific and age-specific incidence rate differences of NCGC using the SEER database and the joinpoint regression software. This study is potentially interesting to understand the epidemiology of NCGC but in its current form, this manuscript is a bit too simplistic in the analysis and should be improved from the following aspects. 

1. Please explicitly clarify how the race/ethnicity information was collected. Is it self-reported? Is there any percentage ancestry information available for NCGC by ADMIXTURE? If so, how would the results change if using the percentage ancestry metrics? And please report the number of patients per category (race) in figure 1 other than only the incidence rate (and report how many patients refused to disclose race information).

2.  Why use a T-test? A t-test indicates that the data are following a normal distribution. Since this study is mainly about statistical analysis, the authors should first perform a Shapiro-Wilk test to confirm normal distribution before using T-test in every case. 

3. Please fix the axis labels and titles in Figure 1. And include CI in the figures. In Figure 1B and 1D, it is clear that in 2016, the incidence of women caught us with men. Please comment on that. 

4. In Figure2, fix the axis labels and titles. And why are Figure 2B,C,D having two regression lines of the same color? How did the authors compute the parallel trend? Please describe the method in more detail (for example, have the authors taken into consideration the standard deviation and how would that affect the test?). In Figure 2D, there is a shift of trend between 2002 and 2004. What happened there and the authors should discuss this trend change and maybe look back even further before 2000 to see how the trend progressed. 

5. This study adopted a public database and used an established package for analysis. Therefore to strengthen the conclusion and impact of this study, the authors should at least attempt to provide more functional implication as in why the difference of incidence rates are found between men and women, between young and old, and among different racial groups. Are there any established studies about the genetic differences or are there any sex-specific or age-specific mutations/copy number changes? 

Reviewer 3 Report

Oh et al., have highlighted the rising incidence of non-cardia gastric cancer among young women compared to young men, especially in non-Hispanic white women in the United States.

Overall the article is well written, structured and is communicable to the specific audience. However, there are a number of minor issues that need to be addressed to enhance the quality of the article.

1.      The authors have included possible explanation of increasing NCGC in women than men. A more detailed explanation of increase of NCGC in younger women compared to older women (age ≥ 55) should be included in the discussion.

2.      Have the authors also conducted any stratified analysis of height and body weight in younger adults? Have they seen any correlation between increase of NCGC in younger non-Hispanic white women and their height and body weight? A comment on that will be appropriate.

3.      Findings from sex-specific trends by staging at diagnosis should be explained in the discussion.

Reviewer 4 Report

This study is interesting with clinical significance. In this study, the authors supported that NCGC incidence has been increasing at a greater rate in younger women compared to counterpart men and this disproportionate increase was mainly seen in young non-Hispanic White women. The authors put forward a new point of view to investigate for etiologies of these trends. The followings are some comments to the authors.

Comments:

The underlying risk factors of the increasing NCGC incidence at a greater rate in younger women (especially in young non-Hispanic White women) should be studied further. The following research directions are my suggestions. For example, the relationship between H. Pylori infection and NCGC incidence the authors mentioned in Discussion in younger women, especially in young non-Hispanic White women.

The following paper is for the author's reference,

Thrift Aaron P,Wenker Theresa Nguyen,El-Serag Hashem B,Global burden of gastric cancer: epidemiological trends, risk factors, screening and prevention.[J] .Nat Rev Clin Oncol, 2023, undefined: undefined.

Round 2

Reviewer 2 Report

The authors have adequately addressed all previous comments. And clarity of the statistical analysis conducted in the manuscript has greatly improved.